# Analysis of Risk Assessment in a Municipal Wastewater Treatment Plant Located in Upper Silesia

**Magdalena Łój-Pilch** * and **Anita Zakrzewska**

Institute of Water and Wastewater Engineering, Silesian University of Technology, 44-100 Gliwice, Poland; anita.zakrzewska@polsl.pl

*   Correspondence: magdalena.loj-pilch@polsl.pl

**Abstract:** Nowadays, risk management applies to every technical facility, branch of the economy, and industry. Due to the characteristics of the analyzed wastewater treatment plant and the specificity of the used processes, one must approach different areas individually. Municipal sewage treatment plants are technical facilities; they function as enterprises and are elements of larger systems—water distribution and sewage disposal. Due to their strategic importance for the environment and human beings, it is essential that they are covered by risk management systems. The basic stage of risk management is its assessment. On its basis, strategic decisions are made and new solutions are introduced. Constant monitoring of the operation of a treatment plant allows for assessment of whether actions taken are correct and whether they cause deterioration of the quality of sewage. In our work, we present a method of risk assessment based on historical data for an existing facility and obtained results.

**Keywords:** municipal wastewater treatment plant; risk management; risk assessment; risk analysis; biological treatment; chemical treatment

## 1. Introduction

Municipal sewage treatment plants are strategic elements of infrastructure and special technical facilities, whose proper functioning determines environmental cleanliness, as well as, people's health. The individual stages of wastewater treatment use physical, biological, and chemical processes that are interrelated and dependent on each other. The effectiveness of each stage is affected by various negative factors, such as the variable composition of incoming sewage and atmospheric conditions. Operators need to limit the effects of events caused by these factors and even prevent their occurrence [1–3]. Therefore, the proper functioning municipal sewage treatment plant should be supported by a risk management system.

The risk of municipal sewage treatment plants can be examined and analyzed at various stages of the treatment plant operation. Considering the potential risk as early as the design stage of the facility allows for choice of the most appropriate trade-off between costs of measures and risks [4,5]. Currently, in Poland, modernization of existing obsolete objects is more common than emergence of new ones. Correctly carried out modernization should be based on risk analysis [6]. Modernization may involve repairs of existing equipment and improvement of technological conditions, or it may be considered as an extension of the technological line with modern devices, e.g., membranes. In particular, the second case should be preceded by a thorough analysis of costs, losses, and risks [7]. In addition, the risks should be monitored throughout the operation of the treatment plant, and the risk management system should be used effectively [8].

This paper discusses the application of risk assessment procedures for the management of municipal wastewater treatment plants, using a facility in Poland as a case study. The method of risk assessment is presented based on historical data.

Current research concerns individual chemical compounds: pharmaceuticals [9], antibiotics [10], individual devices of a treatment plant's technological line, and assessment of ecological risk of receiver after discharge from a sewage treatment plant [11–13]. In contrast to the cited papers (which are examples of research conducted thus far), we assess the risk associated with the entire wastewater treatment plant, which is a novelty in the scientific literature.

## 2. Theory of Risk Management

Defining the risk management process is difficult due to the multitude of various scientific and economic areas in which it is used. It can be defined as a way to find the most optimal methods for conscious, uninterrupted diagnosis and risk control [14]. Risk management should lead to risk setting at an acceptable level [15]. With regard to municipal wastewater treatment plants, risk management can be defined as preventing an occurrence of undesirable events and reducing the size of resulting damage after such events occur [16]. These actions should be carried out on the basis of continuous monitoring of the treatment plant operation, staff training, maintenance of technological process equipment, and maintenance of technical services.

The risk management process can be divided into two basic stages: risk assessment and risk control [17]. The components of risk assessment are identification, estimation, and determination of its acceptability [16]. Risk control involves the monitoring of sewage treatment plant operation and observation of introduced changes.

### 2.1. Risk Identification

The basic method of risk identification of a municipal sewage treatment plant is analysis of historical data, during which attention should be paid to all events causing damage. Due to the nature of the sewage treatment plant, risk identification should be an ongoing process in order to identify new threats and verify those already recognized [18,19]. The process of risk identification is the basis for risk management, and its correct functioning determines the success of the entire risk management process [20].

### 2.2. Risk Estimation

Risk estimation consists of determining its measure, which is dependent on the availability of data, reliability, and expected results [16,21]. In general, risk estimation methods can be divided into three groups [22]:

- Quantitative methods consist of two defining parameters: frequency of occurrence and value of losses; the results are objective and comparable.
- Qualitative methods include a subjective assessment based on knowledge and experience; the results are presented in a descriptive form.
- Mixed methods are the most commonly used type of strategy, involving the simultaneous use of quantitative and qualitative methods.

In the case of sewage treatment plants, the specificity of the collected data allows only the mixed method to be used. The qualitative method is used to identify risk, while the quantitative method is used in risk assessment, assigning specific values to described events.

The result of the mixed method is the so-called risk map (Figure 1). The risk map gives the possibility of a comprehensive presentation of the identified and quantified risk, but it is also a tool helpful in indicating which methods of risk control will work best for a given risk [23]. The risk map presented in this paper is the simplest type of possible risk matrix, determined by the size of losses and the frequency of their occurrence.

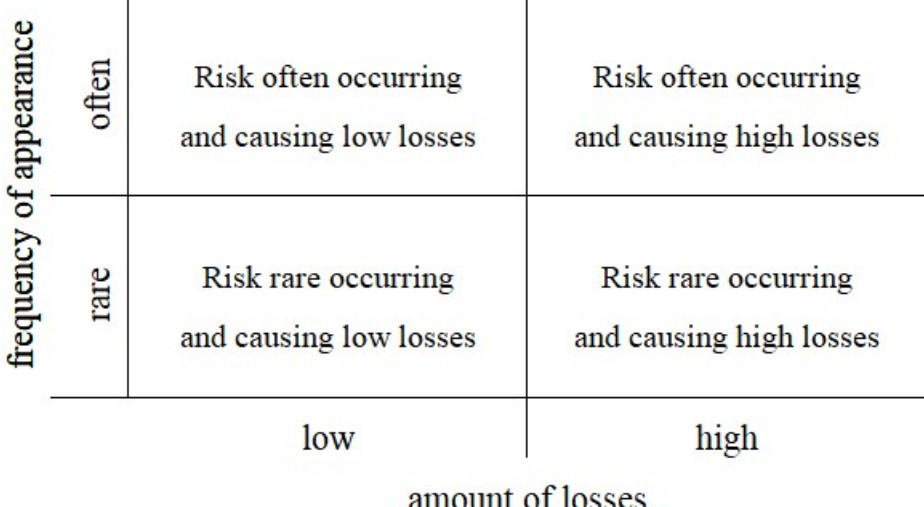

**Figure 1.** Sample scheme of a very simple risk map [19].

*2.3. Risk Admissibility*

In the literature [16,24] and practice, three basic degrees of risk acceptability are distinguished as follows:

- Acceptable risk—an event irrelevant to the general operation of the facility as a "daily risk"; it does not require special security measures.
- Tolerable risk—medium risk, requires intervention, provided that the cost of reducing the risk is reasonable for the damage caused.
- Unacceptable risk—high risk, means an immediate threat to the environment and people and requires immediate steps to limit it.

The degree of admissibility is determined on the basis of legal acts, applicable norms and standards. The Polish legal act for municipal sewage treatment plants is the Regulation of the Minister of Environment from 18 November 2014 [25]. The regulation defines the conditions for discharge of sewage to the receiver. When analyzing a single object, you should also permit considerations for specific water treatment. Graphical interpretations of risk hierarchization (Figure 2) are obtained by applying the acceptability of risk to the risk map (Figure 1) based on the aforementioned documents.

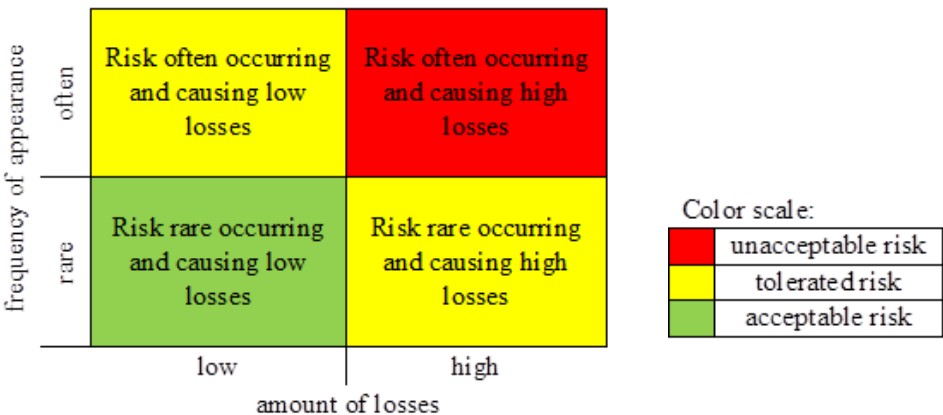

**Figure 2.** Risk hierarchy for the sample scheme presented in Figure 1 [21,22,26,27].

## 3. Obtained Results and Discussion

We analyzed the work of a sewage treatment plant located in Upper Silesia that uses activated sludge technology. The municipal wastewater treatment plant serves 62,000 inhabitants. Based

on historical data collected by the operator of the sewage treatment plant from 2014 to 2016, risk identification and assessment were carried out. Our previous work [28] presents the risk identification that is assessed in the present paper (Table 1). The occurring risk factors (inside, outside, internal, external, latent, explicit) were identified, and the type of risk event (qualitative, operational, ecological, financial) was recognized according to the classification by Iwanejko and Rybicki [16].

**Table 1.** Examples of events together with the incidence rate (I) and the number of losses (L).

| Device | Event | Type of Risk * | Risk Identification [16] | | | Risk Assessment | | |
| | | | Factor ** | Effect | Action Taken/Proposed | Number of Losses(L) | Frequency of Appearance | |
| | | | | | | | (1/year) | (F) |
| sifters | sifter scraper failure | Q | O | clogging of sifter | repair of scrapper | 1 | 0.67 | 1 |
| grit chamber | large dump of greasy sewage | Q | E | clogging of grease chamber outflow | unclogging the outflow | 1 | 0.33 | 1 |
| activated sludge chamber | emergence of filamentous bacteria | Q, OP | I | formation of scum layer | breaking the scum layer and actions aimed at stopping bacteria development | 2 | 13.67 | 3 |
| secondary settling tank | auxiliary devices failure | Q, OP | O | minor disturbance in the settling tank operation | repair of auxiliary devices | 2 | 4.67 | 2 |
| all devises of sewage treatment | electrical power outage | Q, OP, EC, Fi | E | no power for electrical powered devices | connection to emergency power supply | 4 | 0.33 | 1 |

* Q—qualitative, OP—operational, EC—ecological, Fi—financial. ** O—Ordinary, E—external, I—internal.

In the process of risk identification [28], 32 different threats were identified, which occurred 114 times at different frequencies. All of these events were divided according to the frequency of their occurrence, as seen in Table 2, and on the basis of their specific type of risk, a quantitative loss value was assigned to them (Table 3). Results obtained in this way are presented in Table 1. Based on Table 2, a risk map was prepared with the admissibility hierarchy indicated. In order to accurately analyze these events, a risk map was divided into individual devices of the technological line (Figure 3).

**Table 2.** Frequency of appearance (F).

| Occuring Events: | Frequency of Appearance | |
| | (1/year) | (F) |
| rarely | ≤4 or 5 | 0–1 |
| often | 4 or 5–9 | 1–2 |
| very often | ≥9 | 2–3 |

**Table 3.** Number of losses (L).

| Type of Risk | Amount of Losses (L) |
| --- | --- |
| qualitative | 1 |
| qualitative, operational | 2 |
| qualitative, operational, financial | 3 |
| qualitative, operational, ecological, financial | 4 |

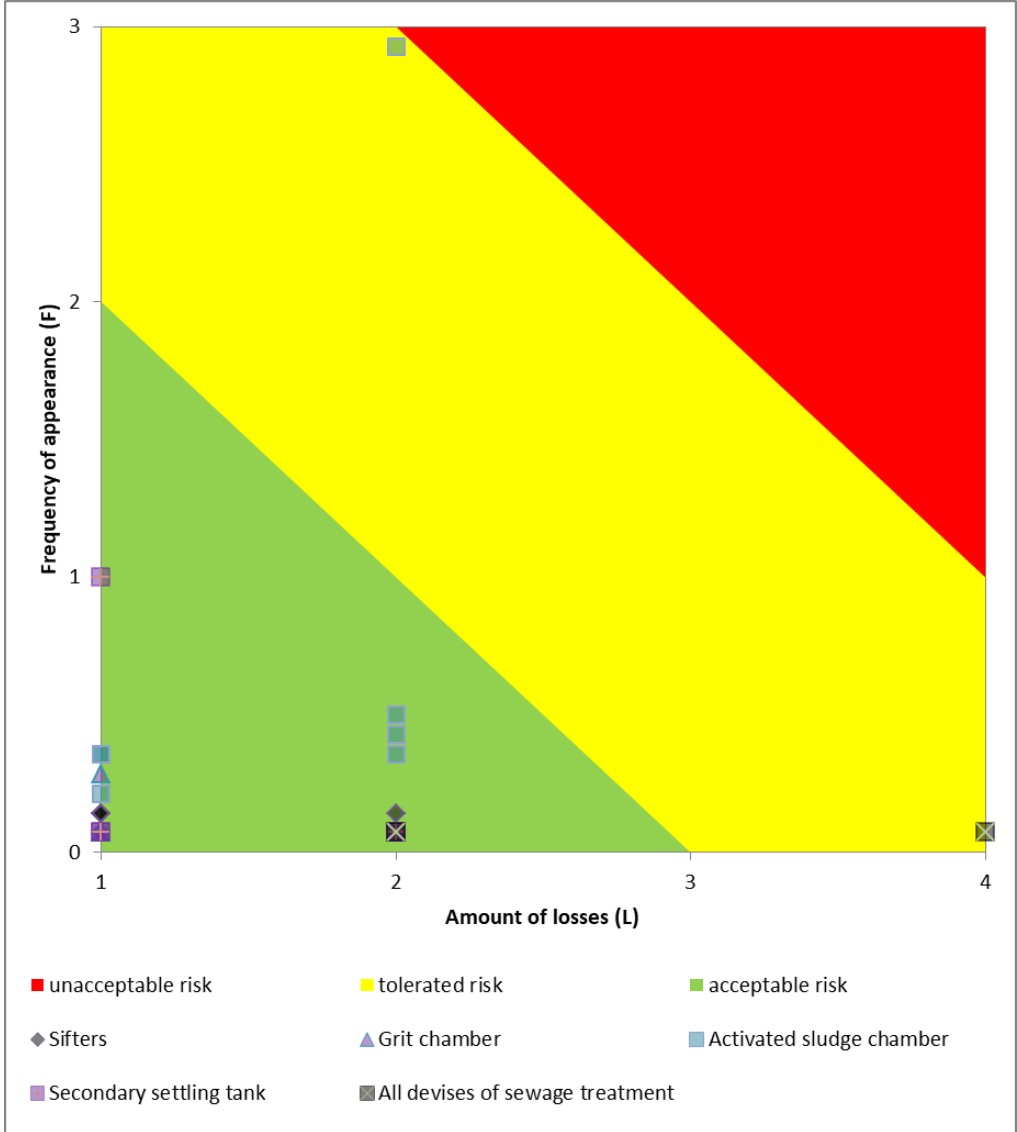

**Figure 3.** The risk map divided into technological line devices, taking into account the risk hierarchy.

The green color in Figure 3 is an acceptable risk area—these are events that do not require a reaction from the operator, and their effects are removed during the normal work of the personnel. The area of tolerated risk is marked in yellow; these events require a response from the staff, but those actions do not have to be taken immediately. The unacceptable risk area is represented by the red color, corresponding to the events that require immediate response from staff and relevant services, regardless of cost. Figure 3 presents 32 events that occurred 114 times. Almost all of the identified hazards are in the area of acceptable risk (30 events that occurred 72 times [28]), which proves the proper functioning of the municipal wastewater treatment plant. Only two events (emergence of filamentous bacteria in an activated sludge chamber, which occurred 41 times, and electrical power

outage, which occurred once [28]), posed a greater threat to operation of the treatment plant, and therefore, they are in the area of tolerated risk.

The activated sludge chamber, where a tolerated risk event occurred, is a technological device, responsible for biological wastewater treatment. It works under variable loads of pollutant and under conditions of variable hydraulic loads. This device should be under strict control and supervision. The identified disruption of work in the activated sludge chamber was caused by the emergence of filamentous bacteria. The bacteria, analyzed in 2015, often appeared due to attempts to improve working conditions in the chamber and the testing of new technological solutions. This is an example of an attempt to modernize, which was not preceded by a risk analysis and gave the operator more problems than benefits.

Another event where the risk was tolerated was for the whole treatment plant. The recorded event concluded in a power failure to the entire facility. Such an event causes a great threat to the proper functioning of the treatment plant and may led to environmental contamination. In the case of the analyzed treatment plant, this did not occur because the facility was equipped with a power generator, and during the failure, strategic devices of the process line were working.

We conducted a similar study for another sewage treatment plant [27] (SWT-2). In comparison with the treatment plant presented in this paper (SWT-1), there were bar screens instead of sifters, and SWT-2 carried out a chemical dephosphatation process that does not occur at SWT-1. No events for sifters were classified as a tolerated risk for bar screens; there were two such events that occurred 14 times (large fat and meat dump that occurred once and clogging of bars that occurred 13 times [27,28]). In both cases, one event in the activated sludge chamber was classified as a tolerated risk. In SWT-1, it was the emergence of filamentous bacteria that occurred frequently (41 times) with a small number of losses (2). While in SWT-2, it was a problem with the agitators and aeration rotors that occurred once but with a very high number of losses (4) [27]. Thus, events with very different frequencies and different numbers of losses may have a similar level of risk.

For the process of chemical dephosphatation in SWT-2, one event occurred 14 times: sludge floated on the surface of the dephosphatation chamber [27,28]. This did not happen in SWT-1 because there was no dephosphatation chamber in the technological line.

In the case of SWT-2, there were also two events classified as tolerated risk—the dump of greasy wastewater in the grit chamber and auxiliary device failure of the clarifier [27,28]. The dumping of greasy wastewater into the grit chamber was also reported in SWT-1, but in this case, it was classified as an acceptable risk because it occurred more frequently (once in SWT-1 but eight times in SWT-2 [27,28]). The auxiliary device failure of the clarifier did not occur in SWT-1. Based on these analyses, it can be concluded that SWT-1 is more reliable than SWT-2.

## 4. Conclusions

The analyzed municipal sewage treatment plant functions properly—none of the identified threats were classified as an unacceptable risk area. In addition, only two events were in the area of tolerated risk. The remaining 112 irregularities that occurred in the three-year period analyzed were events of acceptable risk—everyday risk. These are minor irregularities in the operation of individual devices that a well-trained crew can easily handle in the course of normal operations.

The proposed risk assessment method is only adequate for sites that have complete and good-quality historical data (detailed, consistently described, regularly collected), because it is based on risk identification. Assigned, on the basis of previously performed identification, weights for individual frequency of occurrence and the start size, determined on the basis of the type of risk, were correctly selected, which we concluded after identifying an adequate events distribution on the risk map (Figure 3).

Research to date has focused on risks associated with environmental pollution with chemicals and their impact on the functioning biological part of a sewage treatment plant and on the effects of discharge of such treated wastewater to receivers (rivers) [9–13]. In the framework of a larger

project, this article presents only a fragment of research, the purpose of which is to look at sewage treatment plants as one organism. Based on the results obtained, the next stage of research is to develop appropriate weights for individual technological line devices. They will be assigned to individual devices based on their impact on the quality of treatment plant operations. These weights are necessary to define strategies to minimize risks and to prepare the risk management procedures in sewage treatment plants. The introduction of procedures, which are going to be developed, will facilitate the management of municipal wastewater treatment plants. Currently, there is a lack of unified procedures for managing risk at sewage treatment plants.

**Author Contributions:** Conceptualization and methodology M.Ł.-P. and A.Z.; formal analysis, writing—original draft preparation, M.Ł.-P.; writing—review and editing M.Ł.-P. and A.Z. All authors have read and agreed to the published version of the manuscript.

**Funding:** This work was supported by Research Funds for Young Researchers, awarded to the Institute of Water and Wastewater Engineering of the Silesian University of Technology.

**Conflicts of Interest:** The authors declare no conflict of interest.

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
