# Peer review of "Analysis of Risk Assessment in a Municipal Wastewater Treatment Plant Located in Upper Silesia"

_water, doi:10.3390/w12010023_

Round 1

Reviewer 1 Report

There are no grammatical errors.

Author Response

Dear Reviewer,
Thank you for your revision of our article.

Reviewer 2 Report

 Analysis of risk assessment in municipal wastewater treatment plant on example of object located in Upper Silesia

The paper addresses a methodology for risk assessment of a wastewater treatment plan. The paper has no scientific relevance since is just a simple application with historical data.

In which way can this approach help the decision-makers to, in advance, define strategies to minimize risks.

The paper also lacks in comparing the results with others methodologies and in that way clarify the benefits of this approach.

Throughout the text, I also have the following comments:

Line 42 – it is not clear, the paper is about risk management or assessment?

Line 51 to 53 – Authors only addressed risk assessment, what about risk control? Together with the previous comment and the title, it becomes a little confusing what is the scope of the paper.

Table 1 – “Sewage treatment” as a device? Usually, sewage treatment has several devices.

Line 78 – Figure has a typo. And all the other figure legends. Legends are overall very poor, the legends do not explain the figure. In this case the “risk matrix – risk map” is difficult to understand.

Line 95 – the figure could be merged with figure 1. It is not a correct scientific journal type of writing or presenting the information.

Line 107 - losses (L) but in Line 109 Losses (S)

Table 4 – possibility to merge with figure 1, half the table is the same.

Figure 3 – are all the 3 events or the 114 times presented in the Figure? Which events are those?

Figure 4 – The same situations, the only difference to figure 3 is that the events are discriminated.

Very poor discussion of the results and conclusions, the paper has no novelty at all.

After I had undertaken a thorough reading on this version of the manuscript is my suggestion that the manuscript is not suitable to be published.

Author Response

Dear Reviewer,
Thank you for your valuable comments. We have improved our paper in accordance with them. We do belive the paper gained in value thanks to your suggestions.

1) The paper addresses a methodology for risk assessment of a wastewater treatment plan. The paper has no scientific relevance since is just a simple application with historical data.

We do believe the paper provide useful insights on how to use historical data for risk assessment methods, as it was also pointed by other reviewers. The historical data allow to assess whether taken actions have been correct and to draw conclusions for future exploatation. 

2)In which way can this approach help the decision-makers to, in advance, define strategies to minimize risks.

"Based on the results obtained, the next stage of research is needed to develop appropriate weights for individual technological line devices. These weights are necessary to define strategies to minimize risks and to prepare the risk management procedures in sewage treatment plants." This explanation have been added to article in 4. Conclusions. Thank you for paying attention to this aspect. 

3)The paper also lacks in comparing the results with others methodologies and in that way clarify the benefits of this approach.

In the case of sewage treatment plant, the specificity of the collected data allows only the mixed method to be used. This explanation has been added in article (line 75).

4) Line 42 – it is not clear, the paper is about risk management or assessment?

This paper presents the method of risk assessment based on historical data for an existing facility and obtained results. The risk assessment is one of two stages in the risk management, that's why in authors opinion the theory of risk management should be included in the theoretical part of work. The title of the second unit has been changed to "2. Theory of risk management"

5) Line 51 to 53 – Authors only addressed risk assessment, what about risk control? Together with the previous comment and the title, it becomes a little confusing what is the scope of the paper.

As explained in the previous comment and in 1. Introduction (line 42), the paper presents the method of risk assessment. Risk control together with risk assessment are stages of risk management. That's why authors define them in the theoretical part "2. Theory of risk management". However, the risk control is not the aim of this work. The risk control may be the future part of the research. 

6) Table 1 – “Sewage treatment” as a device? Usually, sewage treatment has several devices.

The term "sewage treatment" in Table 1 has been changed into "all devises of sewage treatment". 

7) Line 78 – Figure has a typo. And all the other figure legends. Legends are overall very poor, the legends do not explain the figure. In this case the “risk matrix – risk map” is difficult to understand.

The figures and all the other figure legends have been improved. The description of risk matrix and risk map" has been added (line 73): "The result of the mixed method is the so-called risk map (Figure 1). The risk map gives the possibility of a comprehensive presentation of the identified and quantified risk, but it is also a tool helpful in indicating which methods of risk control will work best for a given risk. The presented in this paper risk map is the simplest type of possible risk matrix, determined on the basis of the size of losses and the frequency of their occurrence."

8) Line 95 – the figure could be merged with figure 1. It is not a correct scientific journal type of writing or presenting the information.

The figure 2. presents the risk hierarchy for sample scheme presented in Figure 1. This description has been added to the Figure 2 legend. 

9) Line 107 - losses (L) but in Line 109 Losses (S)

This typo has been corrected.

10) Table 4 – possibility to merge with figure 1, half the table is the same.

Table 4 has been merged with Table 1. 

11) Figure 3 – are all the 3 events or the 114 times presented in the Figure? Which events are those?

The figure 3 presents 32 events that occurred 114 times. Almost all of the identified hazards are in the area of ​​acceptable risk (30 events that occurred 72 times). Two events (emergence of filamentous bacteria in activated sludge chamber that occurred 41 times and electrical power outage that occurred once) are in area of ​​tolerated risk.

12) Figure 4 – The same situations, the only difference to figure 3 is that the events are discriminated.

Figure 4 has been merged with the figure 3.

13) Very poor discussion of the results and conclusions, the paper has no novelty at all.

There are no detailed research on this subject in sewage treatment plants, what we believe, make our research novelty.

We have improved the results discussion and reffered to the authors' papers published before:

DOI: 10.1051/e3sconf/201910000050

and 

https://www.taylorfrancis.com/books/e/9781351174664/chapters/10.1201/9781351174664-242

"The similar study was conducted by authors for another sewage treatment plant  (SWT-2). In comparison with the treatment plant presented in this paper (SWT-1), there were bar screen instead of sifters and SWT-2 carries out a chemical dephosphatation process that does not occur on SWT-1. No event for sifters were classified as tolerated risk, when for bar screen there were 2 such events that occurred in total 14 times (large fat and meat dump that occurred once and clogging of bars that occurred 13 times). In both cases one event in activated sludge chamber was classified as tolerated risk. In SWT-1 it was emergence of filamentous bacteria that occurred frequent (41 times) with small amount of losses (2), while in SWT-2 it was problem with the agitators and aeration rotors that occurred once but with very high amount of losses (4). Thus, events with very different frequencies and different amount of loss may have a similar level of risk.

For the process of chemical dephosphatation in SWT-2, 14 times was noticed one event, sludge floated on the surface of dephosphatation chamber. It didn’t happen in SWT-1 because there is no dephosphatation chamber in technological line.

In the case of SWT-2 there were also 2 events classified as tolerated risk - dump of greasy wastewater in grit chamber and auxiliary devices failure of clariffer. Dump of greasy wastewater in grit chamber was noticed also in SWT-1, but in this case it was classified as acceptable risk because it occurred more frequently (once in SWT-1 while eight times in SWT-2). The auxiliary devices failure of clariffer did not occur in SWT-1. Based on these analysis, it can be concluded that SWT-1 is more reliable than that SWT-2."

14) After I had undertaken a thorough reading on this version of the manuscript is my suggestion that the manuscript is not suitable to be published.

We have improved our paper in accordance with them. We do belive the paper gained in value thanks to your suggestions.

Reviewer 3 Report

This paper discusses the application of risk assessment procedures for the management of municipal wastewater treatment plants, using a facility in Poland as case-study.

While providing useful insights on how to use historical data for risk assessment methods, there several issues which need to be addressed. These are detailed in the following sections.

Regarding English language and style several spell checks and grammar corrections are required (lines: 57, 72, 74, 77, 81, 98, 114, 121, 126). There is no consistency in the use of third singular person or first plural person throughout the manuscript. Several passages contain sentences which are too fragmented (Paragraph 2.1; lines 77-78; lines 131-134; Paragraph 2.3) and are not clear. Also, some of the used terminology would require further explanations/description. For example: since the title, authors use the term ‘object’ but what do they mean exactly? a facility? They would need to be more specific. Also, they refer twice to ‘an earlier stage of research’, but is it not clear to what they are exactly referring (i.e. treatment stage? design stage? both?)

Other issues to be addressed:

Line 56: what are the components of risk control? You only define the components of risk assessment

Lines 64:67: Do not leave these key points in brackets. List them clearly, providing more details on their description

Table 1:

At this point in the manuscript there would need more generic events to be described. In an activated sludge chamber there could be many more events causing serious troubles. Why authors focus on this particular one is explained only later in the manuscript. Authors refer to ‘sewage treatment’. I think it would be beneficial to add ‘overall treatment’ or ‘overall plant’, otherwise it is not clear why at this stage they mention the power failure in the event

Line 69: when writing ‘measure’ do authors mean ‘quantification’?

Line 111: define ‘appropriately’

Line 141: define ‘good’ data

Line 146:147: How do authors suggest this would be possible?

Also, what else do authors suggest is needed in future research? How can we facilitate this passage of using risk assessment strategies? How is authors’ research work advancing current knowledge and improving current practices?

Author Response

Dear Reviewer,
Thank you for your valuable comments. We have improved our paper in accordance with them. We do belive the paper gained in value thanks to your suggestions.

1) Regarding English language and style several spell checks and grammar corrections are required (lines: 57, 72, 74, 77, 81, 98, 114, 121, 126). There is no consistency in the use of third singular person or first plural person throughout the manuscript. Several passages contain sentences which are too fragmented (Paragraph 2.1; lines 77-78; lines 131-134; Paragraph 2.3) and are not clear. Also, some of the used terminology would require further explanations/description. For example: since the title, authors use the term ‘object’ but what do they mean exactly? a facility? They would need to be more specific. Also, they refer twice to ‘an earlier stage of research’, but is it not clear to what they are exactly referring (i.e. treatment stage? design stage? both?)

The English language and terminology have been improved.

2) Line 56: what are the components of risk control? You only define the components of risk assessment

The explanation has been added to the 2. Theory of risk management: "The risk control involves monitoring of sewage treatment plant operation and observation of introduced changes." 

3) Lines 64:67: Do not leave these key points in brackets. List them clearly, providing more details on their description

The whole paragraph has been rewritten. The details have been included.

4) Table 1: At this point in the manuscript there would need more generic events to be described. In an activated sludge chamber there could be many more events causing serious troubles. Why authors focus on this particular one is explained only later in the manuscript. Authors refer to ‘sewage treatment’. I think it would be beneficial to add ‘overall treatment’ or ‘overall plant’, otherwise it is not clear why at this stage they mention the power failure in the event

The term "sewage treatment plants" has been changed into "all devises of sewage treatment". According to the comment of Reviewer 2, Table 1 has been merged with Table 4 and renumbered as Table 3.

More genetic event are desribed in authors' paper: 

Łój-Pilch M, Zakrzewska A, Zielewicz E, Impact of human factors on threats in sewage treatment plants, In: Safety and Reliability – Safe Societies in a Changing World; Haugen S, Vinnem JE, Barros A, Kongsvik T, Gulijk C, Taylor & Francis Group, London 2018, pp 1933 – 1938 (https://www.taylorfrancis.com/books/e/9781351174664/chapters/10.1201/9781351174664-242)

We believe that it is not necessary to repeat this information because the mentioned article is both available online and cited in References.

In newly created, table 3 (Table 1 merged with table 4), only example events are presented. Authors decided to present one example event for each device. It shoud be noted that the discussion of results (Figure 3) include all of the events that occured in the tested treatment plans (32 events, that occured 114 times). We believe that there is no need to present all of the identified events, as there are o lot of them and they are desribed in detail in previously mentioned paper, available on-line (https://www.taylorfrancis.com/books/e/9781351174664/chapters/10.1201/9781351174664-242).

5) Line 69: when writing ‘measure’ do authors mean ‘quantification’?

As described in "2.2. Risk estimation", there are 3 methods used to measure risk. One of them is quantitative, but the other one is qualitive. The third one is  mixed. Writing "measure" authors mean "to discover the exact size or amount of something,  to judge the quality, effect, importance, or value of something", acording to Cambridge English Dictionary (https://dictionary.cambridge.org/dictionary/english/measure). 

6) Line 111: define ‘appropriately’

The term 'appropriately' (line 113) has been replaced by 'hierarchy' as it concerns the risk hierarchy presented in Figure 2.

7) Line 141: define ‘good’ data

Writing 'good' data authors mean data that are detailed, consistently described, regularly collected. This explanation has been added in article. 

8) Line 146:147: How do authors suggest this would be possible?

The mentioned weight will be assigned to individual devices on the basis of their impact on the final quality of treatment plant operation. These weights are necessary to define strategies to minimize risks and to prepare the risk management procedures in sewage treatment plants.

9) Also, what else do authors suggest is needed in future research? How can we facilitate this passage of using risk assessment strategies? How is authors’ research work advancing current knowledge and improving current practices?

As mentioned above, the development of weights for an individual device that are going to be the aim of the future research, are necessary to define strategies to minimize risks and to prepare the risk management procedures in sewage treatment plants. The introduction of procedures that are planned to be developed will facilitate the management of municipal wastewater treatment plants. Now there is a lack of unified procedures for managing risk at sewage treatment plants. 

There are no detailed research on this subject in sewage treatment plants, what we believe, make our research novelty.

Reviewer 4 Report

Same content as comments for Authors.

Title : Analysis of risk assessment in municipal wastewater treatment plant on example of object located in Upper Silesia

The author first published a three-page manuscript in the MDPI proceedings on methodology and application.

(Title : Risk Assessment Analysis in a Municipal Wastewater Treatment Plant, Proceedings 2019, 16, 18; doi: 10.3390 / proceedings2019016018)

And the author has already published the following manuscript on the web, consisting of seven pages at EDP Sciences' E3S Web of Conferences is an Open Access publication.

(Title : Risk assessment in municipal wastewater treatment plant. E3S Web of Conferences https://doi.org/10.1051/e3sconf / 2019 (100 1000 00 00 EKO-DOK 2019 50 5)

The manuscript requested for review has the same methodology and content (7 pages) as the manuscript presented at the E3S conference and only one reference number has been added.

However, only the data used for risk assessment of some of the amendments is different and there is no difference.

(note: E3S web of conference policy on re-use: Conference papers may be subsequently updated, or enhanced, for further publication as a regular journal article.)

This manuscript has not been further reinforced with methodology and logic from previous research, and no progress has been found.

From my point of view, I recommend re-submission by reinforcing the manuscript.

Regards,

Author Response

Dear Reviewer,
Thank you for your valuable comments. We have improved our paper in accordance with them. We do belive the paper gained in value thanks to your suggestions.

1) The author first published a three-page manuscript in the MDPI proceedings on methodology and application. (Title : Risk Assessment Analysis in a Municipal Wastewater Treatment Plant, Proceedings 2019, 16, 18; doi: 10.3390 / proceedings2019016018)

The mentioned paper (DOI: 10.3390 / proceedings2019016018) belongs to the Proceedings, 2019, ISMO 2019.  This volume of Proceedings aims to gather the papers presented at the Eighth International Conference ISMO’19—Innovations-Sustainability-Modernity-Openness, held on 22–23 May 2019 in Bialystok, Poland. 

The paper requested for review is going to belong to Water Special Issue "Innovations–Sustainability–Modernity–Openness in Water Research", which applies to the same conference (Eighth International Conference ISMO’19—Innovations-Sustainability-Modernity-Openness, held on 22–23 May 2019 in Bialystok, Poland). 

The idea presented by the Conference Organizers was to prepare two papers on the basis of results presented during above mentioned conference. One of the paper has been the proceeding abstract, published in Proceedings as the kind of on-line version of Book of Abstracts and the research selected by the Conference Comitee are requested for the review to Water Special Issue "Innovations–Sustainability–Modernity–Openness in Water Research" in the extended form.

2) And the author has already published the following manuscript on the web, consisting of seven pages at EDP Sciences' E3S Web of Conferences is an Open Access publication. (Title : Risk assessment in municipal wastewater treatment plant. E3S Web of Conferences https://doi.org/10.1051/e3sconf / 2019 (100 1000 00 00 EKO-DOK 2019 50 5). The manuscript requested for review has the same methodology and content (7 pages) as the manuscript presented at the E3S conference and only one reference number has been added. However, only the data used for risk assessment of some of the amendments is different and there is no difference. (note: E3S web of conference policy on re-use: Conference papers may be subsequently updated, or enhanced, for further publication as a regular journal article.)

The mentioned paper (DOI: 10.1051/e3sconf/201910000050) has been prepared parallel in time to the preparation of the article requested for the review (EKO DOK Conference was held in April 2019, while IŚMO Conference was held in May 2019). These papers apply to research carried out in two different sewage treatment plants. The presented results in both articles are intermediate results of a larger research project that is being implemented. The same methodology was used to compare the final results of the entire research project.

According to Reviewer's suggestion, we have reffered to the paper published in E3S by comparing the results obtained in this stage of work in two different sewage treatment plants. 

3) This manuscript has not been further reinforced with methodology and logic from previous research, and no progress has been found.

We have reffered to the papers published before:

"The similar study was conducted by authors for another sewage treatment plant  (SWT-2). In comparison with the treatment plant presented in this paper (SWT-1), there were bar screen instead of sifters and SWT-2 carries out a chemical dephosphatation process that does not occur on SWT-1. No event for sifters were classified as tolerated risk, when for bar screen there were 2 such events that occurred in total 14 times (large fat and meat dump that occurred once and clogging of bars that occurred 13 times). In both cases one event in activated sludge chamber was classified as tolerated risk. In SWT-1 it was emergence of filamentous bacteria that occurred frequent (41 times) with small amount of losses (2), while in SWT-2 it was problem with the agitators and aeration rotors that occurred once but with very high amount of losses (4). Thus, events with very different frequencies and different amount of loss may have a similar level of risk.

For the process of chemical dephosphatation in SWT-2, 14 times was noticed one event, sludge floated on the surface of dephosphatation chamber. It didn’t happen in SWT-1 because there is no dephosphatation chamber in technological line.

In the case of SWT-2 there were also 2 events classified as tolerated risk - dump of greasy wastewater in grit chamber and auxiliary devices failure of clariffer. Dump of greasy wastewater in grit chamber was noticed also in SWT-1, but in this case it was classified as acceptable risk because it occurred more frequently (once in SWT-1 while eight times in SWT-2). The auxiliary devices failure of clariffer did not occur in SWT-1. Based on these analysis, it can be concluded that SWT-1 is more reliable than that SWT-2."

Round 2

Reviewer 2 Report

The paper is still lacking in scientific soundness. The author could not yet demonstrate the potential of this approach to other WWTP risk analysis.

The discussion is still weak. The text introduced between lines 147 and 165 seems out of other paper, since it relates to other case studies and again not demonstrate the novelty of the article.

Author Response

Dear Reviewer,
Thank you for your valuable comments. We have improved our paper in accordance with them. We do belive the paper gained in value thanks to your suggestions.

1) The paper is still lacking in scientific soundness. The author could not yet demonstrate the potential of this approach to other WWTP risk analysis.

Thank you for paying attention to this aspect of the work. According to your suggestion, we referred to research conducted so far.

We added in "1. Introduction": "Current research concerns individual chemical compounds: pharmaceuticals, antibiotics and individual devices of the treatment plant technological line or assessment of ecological risk of receiver after discharge from sewage treatment plant. In contrast to the cited papers (which are examples of research conducted so far), the authors assess the risk associated with the entire wastewater treatment plant, which is novelty in the scientific literature."

2) The discussion is still weak. The text introduced between lines 147 and 165 seems out of other paper, since it relates to other case studies and again not demonstrate the novelty of the article.

Thank you for paying attention to this aspect of the work. According to your suggestion, we referred to research conducted so far.

We added in "4. Conclusions": "Research to date has focused on risks associated with environmental pollution with chemicals and their impact on the functioning of biological part of sewage treatment plant and on effects of discharge of such treated wastewater to receivers (rivers) [ 9-13]. The authors of the work in the framework of a larger project, article presents only a fragment of research, the purpose of which is to look at sewage treatment plants as one organism. "

We do belive the paper gained in value thanks to your suggestions.

Reviewer 4 Report

I fully understand the situation of the previously published papers and the necessity of this manuscript. Unlike the paper SWT-2 published earlier, the author's target sewage treatment plant (SWT-1) has a bar screen instead of shifters, and the treatment process such as chemical phosphorus removal has some other characteristics. Author explained that different conditions (such as failure) and more reliable conclusions can be drawn. However, I still do not believe that this manuscript is more advanced than the previous manuscript only because the results are more reliable, mainly considering only the characteristics of the another sewage treatment plant. Therefore, I still cannot find methodologically advanced parts of the risk assessment analysis technique in this manuscript.

 There may be a mistake in my judgment even though the author has added a meaningful value as a sufficient paper in the calculation of specific coefficients and weighting methods applied in this paper. So I expect wise and objective judgment from other reviewers and editors

Author Response

Dear Reviewer,
Thank you for your valuable comments. We have improved our paper in accordance with them. We do belive the paper gained in value thanks to your suggestions.

I fully understand the situation of the previously published papers and the necessity of this manuscript. Unlike the paper SWT-2 published earlier, the author's target sewage treatment plant (SWT-1) has a bar screen instead of shifters, and the treatment process such as chemical phosphorus removal has some other characteristics. Author explained that different conditions (such as failure) and more reliable conclusions can be drawn. However, I still do not believe that this manuscript is more advanced than the previous manuscript only because the results are more reliable, mainly considering only the characteristics of the another sewage treatment plant. Therefore, I still cannot find methodologically advanced parts of the risk assessment analysis technique in this manuscript.

 There may be a mistake in my judgment even though the author has added a meaningful value as a sufficient paper in the calculation of specific coefficients and weighting methods applied in this paper. So I expect wise and objective judgment from other reviewers and editors

In relation to the suggestions received, the authors added the following text:

in "1. Introduction": "Current research concerns individual chemical compounds: pharmaceuticals, antibiotics and individual devices of the treatment plant technological line or assessment of ecological risk of receiver after discharge from sewage treatment plant. In contrast to the cited papers (which are examples of research conducted so far), the authors assess the risk associated with the entire wastewater treatment plant, which is novelty in the scientific literature."

and in "4. Conclusions": "Research to date has focused on risks associated with environmental pollution with chemicals and their impact on the functioning of biological part of sewage treatment plant and on effects of discharge of such treated wastewater to receivers (rivers) [ 9-13]. The authors of the work in the framework of a larger project, article presents only a fragment of research, the purpose of which is to look at sewage treatment plants as one organism. "

We believe that the article provides useful information on the use of historical data for methods of risk assessment of municipal treated wastewater, which was also mentioned by other reviewers.

We do belive the paper gained in value thanks to your suggestions.